# Domain Adaptation by Using Causal Inference to Predict Invariant Conditional Distributions

**Sara Magliacane**
MIT-IBM Watson AI Lab, IBM Research*
sara.magliacane@gmail.com

**Thijs van Ommen**
University of Amsterdam
thijsvanommen@gmail.com

**Tom Claassen**
Radboud University Nijmegen
tomc@cs.ru.nl

**Stephan Bongers**
University of Amsterdam
srbongers@gmail.com

**Philip Versteeg**
University of Amsterdam
p.j.j.p.versteeg@uva.nl

**Joris M. Mooij**
University of Amsterdam
j.m.mooij@uva.nl

## Abstract

An important goal common to domain adaptation and causal inference is to make accurate predictions when the distributions for the source (or training) domain(s) and target (or test) domain(s) differ. In many cases, these different distributions can be modeled as different contexts of a single underlying system, in which each distribution corresponds to a different perturbation of the system, or in causal terms, an intervention. We focus on a class of such *causal* domain adaptation problems, where data for one or more source domains are given, and the task is to predict the distribution of a certain target variable from measurements of other variables in one or more target domains. We propose an approach for solving these problems that exploits causal inference and does not rely on prior knowledge of the causal graph, the type of interventions or the intervention targets. We demonstrate our approach by evaluating a possible implementation on simulated and real world data.

## 1 Introduction

Predicting unknown values based on observed data is a problem central to many sciences, and well studied in statistics and machine learning. This problem becomes significantly harder if the training and test data do not have the same distribution, for example because they come from different domains. Such a distribution shift can happen whenever the circumstances under which the training data were gathered are different from those for which the predictions are to be made. A rich literature exists on this problem of *domain adaptation*, a particular task in the field of *transfer learning*; see e.g. Quiñonero-Candela et al. [2009], Pan and Yang [2010] for overviews.

When the domain changes, so may the relations between the different variables under consideration. While for some sets of variables $A$, a function $f : A \to Y$ learned in one domain may continue to offer good predictions for $Y \in \mathcal{Y}$ in a different domain, this may not be true of other sets $A'$ of variables. *Causal graphs* [e.g., Pearl, 2009, Spirtes et al., 2000] allow us to reason about this in a principled way when the domains correspond to different external *interventions* on the system, or more generally, to different contexts in which a system has been measured. Knowledge of the causal graph that describes the data generating mechanism, and of which parts of the model are invariant

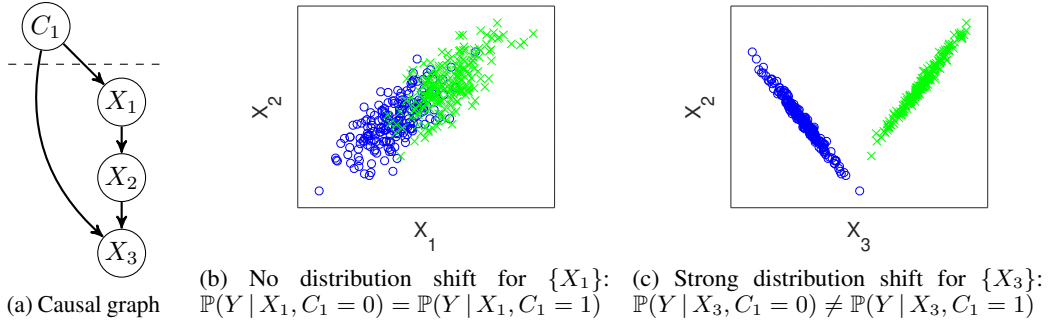

(a) Causal graph

(b) No distribution shift for $\{X_1\}$:
$\mathbb{P}(Y \mid X_1, C_1 = 0) = \mathbb{P}(Y \mid X_1, C_1 = 1)$

(c) Strong distribution shift for $\{X_3\}$:
$\mathbb{P}(Y \mid X_3, C_1 = 0) \neq \mathbb{P}(Y \mid X_3, C_1 = 1)$

Figure 1: In this scenario, an intervention $C_1$ leads to a shift of distribution between source domain and target domain (see also Example 1). Green crosses show source domain data ($C_1 = 0$), blue circles show target domain data ($C_1 = 1$). A standard feature selection method that does not take into account the causal structure, but would use $X_3$ to predict $Y := X_2$ (because $X_3$ is a good predictor of $Y$ in the source domain), would obtain extremely biased predictions in the target domain. Using $X_1$ instead yields less accurate predictions in the source domain, but much more accurate ones in the target domain.

across the different domains, allows one to transfer knowledge from one domain to the other in order to address the problem of domain adaptation [Spirtes et al., 2000, Storkey, 2009, Schölkopf et al., 2012, Bareinboim and Pearl, 2016].

Over the last years, various methods have been proposed to exploit the causal structure of the data generating process in order to address certain domain adaptation problems, each relying on different assumptions. For example, Bareinboim and Pearl [2016] provide theory for identifiability under transfer ("transportability") assuming that the causal graph is known, that interventions are perfect, and that the intervention targets are known. Hyttinen et al. [2015] also assume perfect interventions with known targets but do not rely on complete knowledge of the causal graph, instead inferring the relevant aspects of it from the data. Rojas-Carulla et al. [2018] make the assumption that if the conditional distribution of the target given some subset of covariates is invariant across different source domains, then this conditional distribution must also be the same in the target domain. The methods proposed in [Schölkopf et al., 2012, Zhang et al., 2013, 2015, Gong et al., 2016] all address challenging settings in which conditional independences that follow from the usual Markov and faithfulness assumptions alone do not suffice to solve the problem, but additional assumptions on the data generating process have to be made.

In this work, we will make no such additional assumptions, and address the setting in which both the causal graph *and* the intervention types and targets may be (partially) unknown. Our contributions are the following. We consider a set of relatively weak assumptions that make the problem well-posed. We propose an approach to solve this class of causal domain adaptation problems that can deal with the presence of latent confounders. The main idea is to select the subset of features $A$ that leads to the best predictions of $Y$ in the source domains, while satisfying *invariance* (i.e., $\mathbb{P}(Y \mid A)$ is the same in the source and target domains). To test whether the invariance condition is satisfied, we apply the recently proposed Joint Causal Inference (JCI) framework [Mooij et al., 2018] to exploit the information provided by multiple domains corresponding to different interventions. The basic idea is as follows. First, a standard feature selection method is applied to source domains data to find sets of features that are predictive of a target variable, trading off bias and variance, but unaware of changes in the distribution across domains. A causal inference method then draws conclusions from all given data about the possible causal graphs, avoiding sets of features for which the predictions would not transfer to the target domains. We propose a proof-of-concept implementation of our approach building on a causal discovery algorithm by Hyttinen et al. [2014]. We evaluate the method on synthetic data and a real-world example.

## 2 Theory

Before giving a precise definition of the class of domain adaptation problems that we consider in this work, we begin with a motivating example.

**Example 1.** *We are given three variables $X_1, X_2, X_3$ describing different aspects of a system (for example, certain blood cell phenotypes in mice). We have observational measurements of these three variables (the source domain, designated with $C_1 = 0$), and in addition, measurements of $X_1$ and $X_3$ under an intervention (the target domain, designated with $C_1 = 1$), e.g., in which the mice have been exposed to a certain drug. The domain adaptation task is to predict the values of $Y := X_2$ in the interventional target domain (i.e., when $C_1 = 1$). Let us assume for this example that the causal graph in Figure 1a applies, i.e., we assume that $X_2$ is affected by $X_1$ and affects $X_3$, while $C_1$ affects both $X_1$ and $X_3$ (i.e., the intervention targets the variables $X_1$ and $X_3$). This causal graph implies $\mathbb{P}(Y \mid X_1, C_1 = 0) = \mathbb{P}(Y \mid X_1, C_1 = 1)$. Suppose further that the relation between $X_1$ and $X_2$ is about equally strong as the relation between $X_2$ and $X_3$, but considerably more noisy. Then a feature selection method using only available source domain data, and aiming to select the best subset of features to use for prediction of $Y$ will prefer both $\{X_3\}$ and $\{X_1, X_3\}$ over $\{X_1\}$ (because predicting $Y$ from $X_1$ leads to larger variance than predicting $Y$ from $X_3$, and to a larger bias than predicting $Y$ from both $X_1$ and $X_3$). However, under the intervention ($C_1 = 1$), $\mathbb{P}(Y \mid X_3)$ and $\mathbb{P}(Y \mid X_1, X_3)$ both change,[2] so that using those features to predict $Y$ in the target domain could lead to extreme bias, as illustrated in Figure 1c. Because the conditional distribution of $Y$ given $X_1$ is invariant across domains, as illustrated in Figure 1b, predictions of $Y$ based only on $X_1$ can be safely transferred to the target domain.*

This example provides an instance of a domain adaptation problem where feature selection methods that do not take into account the causal structure would pick a set of features that does not generalize to the target domain, and may lead to arbitrarily bad predictions (even asymptotically, as the number of data points tends to infinity). On the other hand, correctly taking into account the causal structure and the possible distribution shift from source to target domain allows to upper bound the prediction error in the target domain, as we will see in Section 2.3.

## 2.1  Problem Setting

We now formalize the domain adaptation problems that we address in this paper. We will make use of the terminology of the recently proposed Joint Causal Inference (JCI) framework [Mooij et al., 2018].

Let us consider a system of interest described by a set of *system variables* $\{X_j\}_{j \in \mathcal{J}}$. In addition, we model the domain in which the system has been measured by *context variables* $\{C_i\}_{i \in \mathcal{I}}$ (we will use "context" as a synonym for "domain"). We will denote the tuple of all system and context variables as $\boldsymbol{V} = ((X_j)_{j \in \mathcal{J}}, (C_i)_{i \in \mathcal{I}})$. System and context variables can be discrete or continuous. As a concrete example, the system of interest could be a mouse. The system variables could be blood cell phenotypes such as the concentration of red blood cells, the concentration of white blood cells, and the mean red blood cell volume. The context variables could indicate for example whether a certain gene has been knocked out, the dosage of a certain drug administered to the mice, the age and gender of the mice, or the lab in which the measurements were done. The important underlying assumption is that context variables are *exogenous* to the system, whereas system variables are *endogenous*. The interventions are not limited to the perfect ("surgical") interventions modeled by the do-operator of Pearl [2009], but can also be other types of interventions such as mechanism changes [Tian and Pearl, 2001], soft interventions [Markowetz et al., 2005], fat-hand interventions [Eaton and Murphy, 2007], activity interventions [Mooij and Heskes, 2013], and stochastic versions of all these. Knowledge of the intervention *targets* is not necessary (but is certainly helpful). For example, administering a drug to the mice may have a direct causal effect on an unknown subset of the system variables, but we can simply model it as a binary exogenous variable (indicating whether or not the drug was administered) or a continuous exogenous variable (describing the dosage of the administered drug) without specifying in advance on which variables it has a direct effect. We can now formally state the domain adaptation task that we address in this work:

**Task 1** (Domain Adaptation Task). *We are given data for a single or for multiple source domains, in each of which $C_1 = 0$, and for a single or for multiple target domains, in each of which $C_1 = 1$. Assume the source domains data is complete (i.e., no missing values), and the target domains data is complete with the exception of all values of a certain target variable $Y = X_j$. The task is to predict these missing values of the target variable $Y$ given the available source and target domains data.*

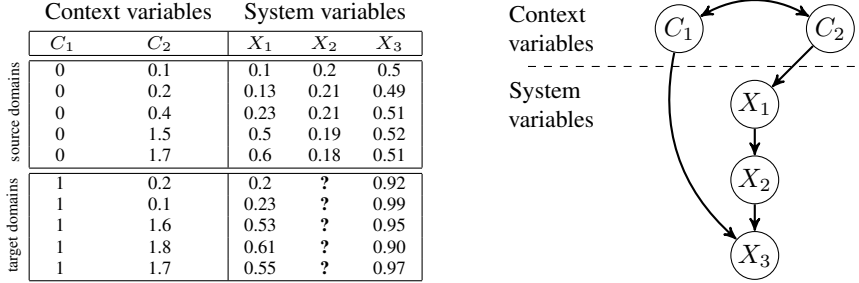

| | Context variables | | System variables | | |
|---|---|---|---|---|---|
| | $C_1$ | $C_2$ | $X_1$ | $X_2$ | $X_3$ |
| source domains | 0 | 0.1 | 0.1 | 0.2 | 0.5 |
| | 0 | 0.2 | 0.13 | 0.21 | 0.49 |
| | 0 | 0.4 | 0.23 | 0.21 | 0.51 |
| | 0 | 1.5 | 0.5 | 0.19 | 0.52 |
| | 0 | 1.7 | 0.6 | 0.18 | 0.51 |
| target domains | 1 | 0.2 | 0.2 | ? | 0.92 |
| | 1 | 0.1 | 0.23 | ? | 0.99 |
| | 1 | 1.6 | 0.53 | ? | 0.95 |
| | 1 | 1.8 | 0.61 | ? | 0.90 |
| | 1 | 1.7 | 0.55 | ? | 0.97 |

Figure 2: Example of a causal domain adaptation problem. The causal graph is depicted on the right, the corresponding data on the left. The task is to predict the missing values of $Y = X_2$ in the target domains ($C_1 = 1$), based on the observed data from the source domains and the target domains, without knowledge of the causal graph. See also Example 2.

An example is provided in Figure 2. In the next subsection, we will formalize our assumptions to turn this task into a well-posed problem.

## 2.2 Assumptions

Our first main assumption is that the data generating process (on both system and context variables) can be represented as a Structural Causal Model (SCM) (see e.g., [Pearl, 2009]):

$$\mathcal{M} : \begin{cases} C_i &= g_i(\boldsymbol{E}_{\text{PA}(i)\cap\mathcal{K}}), & i \in \mathcal{I} \\ X_j &= f_j(\boldsymbol{X}_{\text{PA}(j)\cap\mathcal{J}}, \boldsymbol{C}_{\text{PA}(j)\cap\mathcal{I}}, \boldsymbol{E}_{\text{PA}(j)\cap\mathcal{K}}), & j \in \mathcal{J} \\ p(\boldsymbol{E}) &= \prod_{k\in\mathcal{K}} p(E_k). \end{cases} \quad (1)$$

Here, we introduced exogenous latent independent "noise" variables $(E_k)_{k\in\mathcal{K}}$ that model latent causes of the context and system variables. The parents of each variable are denoted by $\text{PA}(\cdot)$. Each context and system variable is related to its parent variables by a structural equation. In addition, we assume a factorizing probability distribution on the exogenous variables. There could be cyclic dependencies, for example due to feedback loops, but for simplicity of exposition we will discuss only the acyclic case here, noting that the extension to the cyclic case is straightforward given recent theoretical advances on cyclic SCMs [Bongers et al., 2018]. This SCM provides a causal model for the distributions of the various domains, and in particular, it induces a joint distribution $\mathbb{P}(\boldsymbol{V})$ on the context and system variables. Note that we will assume that the data generating process can be modeled by some model of this form, but we do not rely on knowing the precise model.

The SCM $\mathcal{M}$ can be represented graphically by its causal graph $\mathcal{G}(\mathcal{M})$, a graph with nodes $\mathcal{I} \cup \mathcal{J}$ (i.e., the labels of both system and context variables), directed edges $l_1 \to l_2$ for $l_1, l_2 \in \mathcal{I} \cup \mathcal{J}$ iff $l_1 \in \text{PA}(l_2)$, and bidirected edges $l_1 \leftrightarrow l_2$ for $l_1, l_2 \in \mathcal{I} \cup \mathcal{J}$ iff there exists a $k \in \text{PA}(l_1) \cap \text{PA}(l_2) \cap \mathcal{K}$. In the acyclic case, this causal graph is an Acyclic Directed Mixed Graph (ADMG), and $\mathcal{M}$ is also known as a Semi-Markov Causal Model (see e.g., [Pearl, 2009]). The directed edges represent direct causal relationships, and the bidirected edges may represent hidden confounders (both relative to the set of variables in the ADMG). The (causal) *Markov assumption* holds [Richardson, 2003], i.e., any d-separation $\boldsymbol{A} \perp \boldsymbol{B} \mid \boldsymbol{S} \, [\mathcal{G}(\mathcal{M})]$ between sets of random variables $\boldsymbol{A}, \boldsymbol{B}, \boldsymbol{S} \subseteq \boldsymbol{V}$ in the ADMG $\mathcal{G}(\mathcal{M})$ implies a conditional independence $\boldsymbol{A} \perp\!\!\!\perp \boldsymbol{B} \mid \boldsymbol{S} \, [\mathbb{P}(\boldsymbol{V})]$ in the distribution $\mathbb{P}(\boldsymbol{V})$ induced by the SCM $\mathcal{M}$. A standard assumption in causal discovery is that the joint distribution $\mathbb{P}(\boldsymbol{V})$ is *faithful* with respect to the ADMG $\mathcal{G}(\mathcal{M})$, i.e., that there are no other conditional independences in the joint distribution than those implied by d-separation.

We will make the following assumptions on the causal structure (where henceforth we will simply write $\mathcal{G}$ instead of $\mathcal{G}(\mathcal{M})$), which are discussed in detail by Mooij et al. [2018]:

**Assumption 1** (JCI Assumptions). *Let $\mathcal{G}$ be a causal graph with variables $\boldsymbol{V}$ (consisting of system variables $\{X_j\}_{j\in\mathcal{J}}$ and context variables $\{C_i\}_{i\in\mathcal{I}}$).*

*(i) No system variable directly causes any context variable ("exogeneity")*
$(\forall j \in \mathcal{J}, \forall i \in \mathcal{I} : X_j \to C_i \notin \mathcal{G});$

*(ii) No system variable is confounded with a context variable ("randomization")*
$(\forall j \in \mathcal{J}, \forall i \in \mathcal{I} : X_j \leftrightarrow C_i \notin \mathcal{G});$

*(iii) Every pair of context variables is purely confounded ("genericity")*
    *($\forall i, i' \in \mathcal{I} : C_i \leftrightarrow C_{i'} \in \mathcal{G} \land C_i \rightarrow C_{i'} \notin \mathcal{G}$).*

The first assumption is the most crucial one that captures what we mean by "context". The other two assumptions are less crucial and could be omitted, depending on the application. For a more in-depth discussion of these modeling assumptions and on how they compare with other possible causal modeling approaches, we refer the reader to [Mooij et al., 2018]. Any causal discovery method can in principle be used in the JCI setting, but identifiability greatly benefits from taking into account the background knowledge on the causal graph from Assumption 1.

In addition, in order to be able to address the causal domain adaptation task, we will assume:

**Assumption 2.** *Let $\mathcal{G}$ be a causal graph with variables $\boldsymbol{V}$ (consisting of system variables $\{X_j\}_{j \in \mathcal{J}}$ and context variables $\{C_i\}_{i \in \mathcal{I}}$), and $\mathbb{P}(\boldsymbol{V})$ be the corresponding distribution on $\boldsymbol{V}$. Let $C_1$ be the source/target domains indicator and $Y = X_j$ the target variable.*

*(i) The distribution $\mathbb{P}(\boldsymbol{V})$ is Markov and faithful w.r.t. $\mathcal{G}$;*
*(ii) Any conditional independence involving $Y$ in the source domains also holds in the target domains, i.e., if $\boldsymbol{A} \cup \boldsymbol{B} \cup \boldsymbol{S}$ contains $Y$ but not $C_1$ then:[3]*

$$\boldsymbol{A} \perp\!\!\!\perp \boldsymbol{B} \mid \boldsymbol{S} \, [C_1 = 0] \implies \boldsymbol{A} \perp\!\!\!\perp \boldsymbol{B} \mid \boldsymbol{S} \, [C_1 = 1];$$

*(iii) $C_1$ has no direct effect on $Y$ w.r.t. $\boldsymbol{V}$, i.e., $C_1 \rightarrow Y \notin \mathcal{G}$.*

The Markov and faithfulness assumptions are standard in constraint-based causal discovery on a single domain; we apply them here on the "meta-system" composed of system and context.

Assumption 2(ii) may seem non-intuitive, but as we show in the Supplementary Material, it follows from more intuitive (but stronger) assumptions, for example if both the pooled source domains distribution $\mathbb{P}(\boldsymbol{V} \mid C_1 = 0)$ and the pooled target domains distribution $\mathbb{P}(\boldsymbol{V} \mid C_1 = 1)$ are Markov and faithful to the subgraph of $\mathcal{G}$ which excludes $C_1$. These stronger assumptions imply that the causal structure (i.e., presence or absence of causal relationships and confounders) of the other variables is invariant when going from source to target domains. Assumption 2(ii) is a weakened version of these more natural assumptions, allowing additional *independences* to hold in the target domains compared to the source domains, e.g., when $C_1$ models a perfect surgical intervention.

Assumption 2(iii) is strong, yet some assumption of that type seems necessary to make the task well-defined. Without any information at all about the target(s) of $C_1$, or the causal mechanism that determines the values of $Y$ in the target domains, predicting the values of $Y$ for the target domains seems generally impossible. Note that the assumption is more likely to be satisfied if the interventions are believed to be precisely targeted, and gets weaker the more relevant system variables are observed.[4]

As one example of a real-world setting in which these assumptions are reasonable, consider a genomics experiment, in which gene expression levels of many different genes are measured in response to knockouts of single genes. Given our present-day understanding of the biology of gene expression, it is very reasonable to assume that the knockout of gene $X_i$ only has a direct effect on the expression level of gene $X_i$ itself. As long as we do not ask to predict the expression level of $X_i$ under a knockout of $X_i$, but only the expression level of other genes $Y = X_j$ with $j \neq i$, Assumption 2(iii) seems justified. It is also reasonable (based on present-day understanding of biology) to expect that a single gene knockout does not change the causal mechanisms in the rest of the system. This justifies Assumption 2(ii) in this setting if one is willing to assume faithfulness.

In the next subsections, we will discuss how these assumptions enable us to address the domain adaptation task.

## 2.3 Separating Sets of Features

Our approach to addressing Task 1 is based on finding a *separating set* $\boldsymbol{A} \subseteq \boldsymbol{V} \setminus \{C_1, Y\}$ of (context and system) variables that satisfies $C_1 \perp\!\!\!\perp Y \mid \boldsymbol{A} \,[\mathcal{G}]$. If such a separating set $\boldsymbol{A}$ can be found, then the distribution of $Y$ conditional on $\boldsymbol{A}$ is *invariant* under transferring from the source domains to the target domains, i.e., $\mathbb{P}(Y \mid \boldsymbol{A}, C_1 = 0) = \mathbb{P}(Y \mid \boldsymbol{A}, C_1 = 1)$. As the former conditional distribution can be estimated from the source domains data, we directly obtain a prediction for the latter, which then enables us to predict the values of $Y$ from the observed values of $\boldsymbol{A}$ in the target domains.[5]

We will now discuss the effect of the choice of $\boldsymbol{A}$ on the quality of the predictions. For simplicity of the exposition, we make use of the squared loss function and look at the asymptotic case, ignoring finite-sample issues. When predicting $Y$ from a subset of features $\boldsymbol{A} \subseteq \boldsymbol{V} \setminus \{Y, C_1\}$ (that may or may not be separating), the optimal predictor is defined as the function $\hat{Y}$ mapping from the range of possible values of $\boldsymbol{A}$ to the range of possible values of $Y$ that minimizes the *target domains risk* $\mathbb{E}\big((Y - \hat{Y}(\boldsymbol{A}))^2 \mid C_1 = 1\big)$, and is given by the conditional expectation (regression function) $\hat{Y}_{\boldsymbol{A}}^1(\boldsymbol{a}) := \mathbb{E}(Y \mid \boldsymbol{A} = \boldsymbol{a}, C_1 = 1)$. Since $Y$ is not observed in the target domains, we cannot directly estimate this regression function from the data.

One approach that is often used in practice is to ignore the difference in distribution between source and target domains, and use instead the predictor $\hat{Y}_{\boldsymbol{A}}^0(\boldsymbol{a}) := \mathbb{E}(Y \mid \boldsymbol{A} = \boldsymbol{a}, C_1 = 0)$, which minimizes the *source domains risk* $\mathbb{E}\big((Y - \hat{Y})^2 \mid C_1 = 0\big)$. This approximation introduces a bias $\hat{Y}_{\boldsymbol{A}}^1 - \hat{Y}_{\boldsymbol{A}}^0$ that we will refer to as the *transfer bias* (when predicting $Y$ from $\boldsymbol{A}$). When ignoring that source domains and target domains have different distributions, any standard machine learning method can be used to predict $Y$ from $\boldsymbol{A}$. As the transfer bias can become arbitrarily large (as we have seen in Example 1), the prediction accuracy of this solution strategy may be arbitrarily bad (even in the infinite-sample limit).

Instead, we propose to only predict $Y$ from $\boldsymbol{A}$ when the set $\boldsymbol{A}$ of features satisfies the following *separating set* property:

$$C_1 \perp\!\!\!\perp Y \mid \boldsymbol{A} \,[\mathcal{G}], \tag{2}$$

i.e., it d-separates $C_1$ from $Y$ in $\mathcal{G}$. By the Markov assumption, this implies $C_1 \perp\!\!\!\perp Y \mid \boldsymbol{A} \,[\mathbb{P}(\boldsymbol{V})]$. In other words (as already mentioned above), for separating sets, the distribution of $Y$ conditional on $\boldsymbol{A}$ is *invariant* under transferring from the source domains to the target domains, i.e., $\mathbb{P}(Y \mid \boldsymbol{A}, C_1 = 0) = \mathbb{P}(Y \mid \boldsymbol{A}, C_1 = 1)$. By virtue of this invariance, regression functions are identical for the source domains and target domains, i.e., $\hat{Y}_{\boldsymbol{A}}^0 = \hat{Y}_{\boldsymbol{A}}^1$, and hence also the source domains and target domains risks are identical when using the predictor $\hat{Y}_{\boldsymbol{A}}^0$:

$$C_1 \perp\!\!\!\perp Y \mid \boldsymbol{A} \,[\mathcal{G}] \implies \mathbb{E}\big((Y - \hat{Y}_{\boldsymbol{A}}^0)^2 \mid C_1 = 1\big) = \mathbb{E}\big((Y - \hat{Y}_{\boldsymbol{A}}^0)^2 \mid C_1 = 0\big). \tag{3}$$

The r.h.s. can be estimated from the source domains data, and the l.h.s. equals the generalization error to the target domains when using the predictor $\hat{Y}_{\boldsymbol{A}}^0$ trained on the source domains (which equals the predictor $\hat{Y}_{\boldsymbol{A}}^1$ that one could obtain if all target domains data, including the values of $Y$, were observed).[6] Although this approach leads to zero transfer bias, it introduces another bias: by using only a subset of the features $\boldsymbol{A}$, rather than *all available* features $\boldsymbol{V} \setminus \{C_1, Y\}$, we may miss relevant information to predict $Y$. We refer to this bias as the *incomplete information bias*, $\hat{Y}_{\boldsymbol{V} \setminus \{Y, C_1\}}^1 - \hat{Y}_{\boldsymbol{A}}^1$.

The total bias when using $\hat{Y}_{\boldsymbol{A}}^0$ to predict $Y$ is the sum of the transfer bias and the incomplete information bias:

$$\underbrace{\hat{Y}_{\boldsymbol{V} \setminus \{Y, C_1\}}^1 - \hat{Y}_{\boldsymbol{A}}^0}_{\text{total bias}} = \underbrace{(\hat{Y}_{\boldsymbol{A}}^1 - \hat{Y}_{\boldsymbol{A}}^0)}_{\text{transfer bias}} + \underbrace{(\hat{Y}_{\boldsymbol{V} \setminus \{Y, C_1\}}^1 - \hat{Y}_{\boldsymbol{A}}^1)}_{\text{incomplete information bias}}.$$

For some problems, one may be better off by simply ignoring the transfer bias and minimizing the incomplete information bias, while for other problems, it is crucial to take the transfer into account to

obtain small generalization errors. In that situation, we could use any subset $\boldsymbol{A}$ for prediction that satisfies the separating set property (2), implying zero transfer bias; obviously, the best predictions are then obtained by selecting a separating subset that also minimizes the source domains risk (i.e., minimizes the incomplete information bias). We conclude that this strategy of selecting a subset $\boldsymbol{A}$ to predict $Y$ may yield an asymptotic guarantee on the prediction error by (3), whereas simply ignoring the shift in distribution may lead to unbounded prediction error, since the transfer bias could be arbitrarily large in the worst case scenario.

## 2.4 Identifiability of Separating Feature Sets

For the strategy of selecting the best separating sets of features as discussed in Section 2.3, we need to find one or more sets $\boldsymbol{A} \subseteq \boldsymbol{V} \setminus \{C_1, Y\}$ that satisfy (2). Of course, the problem is that we cannot directly test this in the data, because the values of $Y$ are missing for $C_1 = 1$. Note that also Assumption 2(ii) cannot be directly used here, because it only applies when $C_1$ is *not* in $\boldsymbol{A} \cup \boldsymbol{B}$. When the causal graph $\mathcal{G}$ is known, it is easy to verify whether (2) holds directly using d-separation. Here we address the more challenging setting in which the causal graph and the targets of the interventions are (partially) unknown.[7] Conceptually, one could estimate a set of possible causal graphs by using a causal discovery algorithm (for example, extending any standard method to deal with the missing conditional independence tests in $C_1 = 1$), and then read off separating sets from these graphs. In practice, it is not necessary to estimate completely these causal graphs: we only need to know enough about them to verify or falsify whether a given set of features separates $C_1$ from $Y$. The following example (with details in the Supplementary Material) illustrates a case where such reasoning allows us to identify a separating set.

**Example 2.** *Assume that Assumptions 1 and 2 hold for two context variables $C_1, C_2$ and three system variables $X_1, X_2, X_3$ with $Y := X_2$. If the following conditional (in)dependences all hold in the source domains:*

$$C_2 \perp\!\!\!\perp X_2 \mid X_1 \; [C_1 = 0], \qquad C_2 \not\perp\!\!\!\perp X_2 \mid \emptyset \; [C_1 = 0], \qquad C_2 \perp\!\!\!\perp X_3 \mid X_2 \; [C_1 = 0], \qquad (4)$$

*then $C_1 \perp\!\!\!\perp X_2 \mid X_1 \; [\mathcal{G}]$, i.e., $\{X_1\}$ is a separating set for $C_1$ and $X_2$. One possible causal graph leading to those (in)dependences is provided in Figure 2 (the others are shown in Figure 1c in the Supplementary Material). For that ADMG, and given enough data, feature selection applied to the source domains data will generically select $\{X_1, X_3\}$ as the optimal set of features for predicting $Y := X_2$, which can lead to an arbitrarily large prediction error. On the other hand, the set $\{X_1\}$ is separating in any ADMG satisfying (4), so using it to predict $Y$ leads to zero transfer bias, and therefore provides a guarantee on the target domains risk (i.e., it provides an upper bound on the optimal target domains risk, which can be estimated from the source domains data).*

Rather than characterizing by hand all possible situations in which a separating set can be identified (like in Example 2), in this work we delegate the causal inference to an automatic theorem prover. Intuitively, the idea is to provide the automatic theorem prover with the conditional (in)dependences that hold in the data, in combination with an encoding of Assumptions 1 and 2 into logical rules, and ask the theorem prover whether it can prove that $C_1 \perp\!\!\!\perp Y \mid \boldsymbol{A}$ holds for a candidate set $\boldsymbol{A}$ from the assumptions and provided conditional (in)dependences. There are three possibilities: either it can prove the query (and then we can proceed to predict $Y$ from $\boldsymbol{A}$ and get an estimate of the target domains risk), or it can disprove the query (and then we know $\boldsymbol{A}$ will generically give predictions that suffer from an arbitrarily large transfer bias), or it can do neither (in which case hopefully another subset $\boldsymbol{A}$ can be found that does provably satisfy (2)).

## 2.5 Algorithm

A simple (brute-force) algorithm that finds the best separating set as described in Section 2.3 is the following. By using a standard feature selection method, produce a ranked list of subsets $\boldsymbol{A} \subseteq \boldsymbol{V} \setminus \{Y, C_1\}$, ordered ascendingly with respect to the empirical source domains risks. Going through this list of subsets (starting with the one with the smallest empirical source domains risk),

test whether the separating set property can be inferred from the data by querying the automated theorem prover. If (2) can be shown to hold, use that subset $A$ for prediction of $Y$ and stop; if not, continue with the next candidate subset $A$ in the list. If no subset satisfies (2), abstain from making a prediction.[8]

An important consequence of Assumption 2(ii) is that it enables us to transfer conditional independence involving the target variable from the source domains to the target domains (proof provided in the Supplementary Material):

**Proposition 1.** *Under Assumption 2,*

$$A \perp\!\!\!\perp B \,|\, S \,[C_1 = 0] \iff A \perp\!\!\!\perp B \,|\, S \cup \{C_1\} \iff A \perp B \,|\, S \cup \{C_1\} \,[\mathcal{G}]$$

*for subsets $A, B, S \subseteq V$ such that their union contains $Y$ but not $C_1$.*

To test the separating set condition (2), we use the approach proposed by Hyttinen et al. [2014], where we simply add the JCI assumptions (Assumption 1) as constraints on the optimization problem, in addition to the domain-adaptation specific assumption that $C_1 \to Y \notin \mathcal{G}$ (Assumption 2(iii)). As inputs we use all directly testable conditional independence test p-values $p_{A \perp\!\!\!\perp B \,|\, S}$ in the pooled data (when $Y \notin A \cup B \cup S$) and all those resulting from Proposition 1 from the source domains data only (if $Y \in A \cup B \cup S$). If background knowledge on intervention targets or the causal graph is available, it can easily be added as well. We use the method proposed by Magliacane et al. [2016] to query for the confidence of whether some statement (e.g., $Y \perp\!\!\!\perp C_1 \,|\, A$) is true or false. The results of Magliacane et al. [2016] show that this approach is sound under oracle inputs, and asymptotically consistent whenever the statistical conditional independence tests used are asymptotically consistent. In other words, in this way the probability of wrongly deciding whether a subset $A$ is a separating set converges to zero as the sample size increases. We chose this approach because it is simple to implement on top of existing open source code.[9] Note that the computational cost quickly increases with the number of variables, limiting the number of variables that can be considered simultaneously.

One remaining issue is how to predict $Y$ when an optimal separating set $A$ has been found. As the distribution of $A$ may shift when transferring from source domains to target domains, this means that there is a *covariate shift* to be taken into account when predicting $Y$. Any method (e.g., least-squares regression) could in principle be used to predict $Y$ from a given set of covariates, but it is advisable to use a prediction method that works well under covariate shift, e.g., [Sugiyama et al., 2008].

## 3 Evaluation

We perform an evaluation on both synthetic data and a real-world dataset based on a causal inference challenge.[10] The latter dataset consists of hematology-related measurements from the International Mouse Phenotyping Consortium (IMPC), which collects measurements of phenotypes of mice with different single-gene knockouts.

In both evaluations we compare a standard feature selection method (which uses Random Forests) with our method that builds on top of it and selects from its output the best separating set. First, we score all possible subsets of features by their out-of-bag score using the implementation of Random Forest Regressor from `scikit-learn` [Pedregosa et al., 2011] with default parameters. For the baseline we then select the best performing subset and predict $Y$. Instead, for our proposed method we try to find a subset of features $A$ that is also a separating set, starting from the subsets with the best scores. To test whether $A$ is a separating set, we use the method described in Section 2.5, using the ASP solver `clingo 4.5.4` [Gebser et al., 2014]. We provide as inputs the independence test results from a partial correlation test with significance level $\alpha = 0.05$ and combine it with the weighting scheme from Magliacane et al. [2016]. We then use the first subset $A$ in the ranked list of predictive sets of features found by the Random Forest method for which the confidence that $C_1 \perp\!\!\!\perp Y \,|\, A$ holds is positive. If there is no set $A$ that satisfies this criterion, then we abstain from making a prediction.

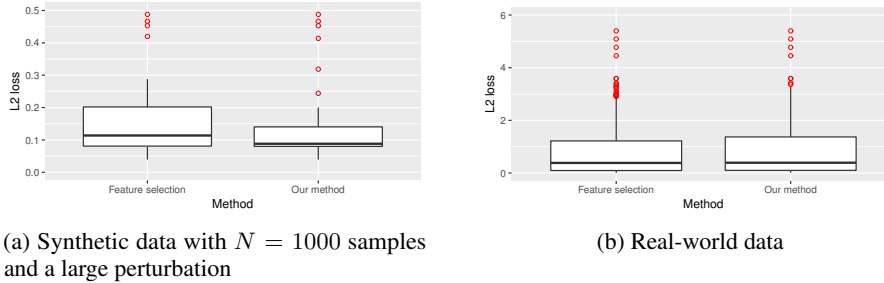

(a) Synthetic data with $N = 1000$ samples and a large perturbation

(b) Real-world data

Figure 3: Evaluation results (see main text and Supplementary Material for details).

For the synthetic data, we generate randomly 200 linear acyclic models with latent variables and Gaussian noise, each with three system variables, and sample $N$ data points each for the observational and two experimental domains, where we simulate soft interventions on randomly selected targets, with different sizes of perturbations. We randomly select which of the two context variables will be $C_1$ and which of the three system variables will be $Y$. We disallow direct effects of $C_1$ on $Y$, and enforce that no intervention can directly affect all variables simultaneously. More details on how the data were simulated are provided in the Supplementary Material. Figure 3a shows a boxplot of the $L_2$ loss of the predicted $Y$ values with respect to the true values for both the baseline and our method, considering the 121 cases out of 200 in which our method does produce an answer. In particular, Figure 3a considers the case of $N = 1000$ samples per regime and interventions that all produce a large perturbation. In the Supplementary Material we show that results improve with more samples, both for the baseline, but even more so for our method, since the quality of the conditional independence tests improves. We also show that, according to expectations, if the target distribution is very similar to the source distributions, i.e., the transfer bias is small, our method does not provide any benefit and seems to perform worse than the baseline. Conversely, the larger the intervention effect, the bigger the advantage of using our method.

For the real-world dataset, we select a subset of the variables considered in the CRM Causal Inference Challenge. Specifically, for simplicity we focus on 16 phenotypes that are not deterministically related to each other. The dataset contains measurements for 441 "wild type" mice and for about 10 "mutant" mice for each of 13 different single gene knockouts. We then generate 1000 datasets by randomly selecting subsets of 3 variables and 2 gene knockout contexts, and always include also "wild type" mice. For each dataset we randomly choose $Y$ and $C_1$, and leave out the observed values of $Y$ for $C_1 = 1$. Figure 3b shows a boxplot of the $L_2$ loss of the predicted $Y$ values with respect to the real values for the baseline and our method. Given the small size of the datasets, this is a very challenging problem. In this case, our method abstains from making a prediction for 170 cases out of 1000 but performs similarly to the baseline on the remaining cases.

## 4 Discussion and Conclusion

We have defined a general class of causal domain adaptation problems and proposed a method that can identify sets of features that lead to transferable predictions. Our assumptions are quite general and in particular do not require the causal graph or the intervention targets to be known. The method gives promising results on simulated data. It is straightforward to extend our method to the cyclic case by making use of the results by Forré and Mooij [2018]. More work remains to be done on the implementation side, for example, scaling up to more variables. Currently, our approach can handle about seven variables on a laptop computer, and with recent advances in exact causal discovery algorithms [e.g., Rantanen et al., 2018], a few more variables would be feasible. For scaling up to dozens of variables, we plan to adapt constraint-based causal discovery algorithms like FCI [Spirtes et al., 2000] to deal with the missing-data aspect of the domain adaptation task. We hope that this work will also inspire further research on the interplay between bias, variance and causality from a statistical learning theory perspective.

**Acknowledgments**

We thank Patrick Forré for proofreading a draft of this work. We thank Renée van Amerongen and Lucas van Eijk for sharing their domain knowledge about the hematology-related measurements from the International Mouse Phenotyping Consortium (IMPC). SM, TC, SB, and PV were supported by NWO, the Netherlands Organization for Scientific Research (VIDI grant 639.072.410). SM was also supported by the Dutch programme COMMIT/ under the Data2Semantics project. TC was also supported by NWO grant 612.001.202 (MoCoCaDi), and EU-FP7 grant agreement n.603016 (MATRICS). TvO and JMM were supported by the European Research Council (ERC) under the European Union's Horizon 2020 research and innovation programme (grant agreement 639466).

## Footnotes

*Most of the work was performed while at the University of Amsterdam.

[2] More precisely, we should say that $\mathbb{P}(Y \mid X_3, C_1 = 1)$ may differ from $\mathbb{P}(Y \mid X_3, C_1 = 0)$, and similarly when conditioning on $\{X_1, X_3\}$.

[3]Here, with $\boldsymbol{A} \perp\!\!\!\perp \boldsymbol{B} \mid \boldsymbol{S} \, [C_1 = 0]$ we mean $\boldsymbol{A} \perp\!\!\!\perp \boldsymbol{B} \mid \boldsymbol{S} \, [\mathbb{P}(\boldsymbol{V} \mid C_1 = 0)]$, i.e., the conditional independence of $\boldsymbol{A}$ from $\boldsymbol{B}$ given $\boldsymbol{S}$ in the mixture of the source domains $\mathbb{P}(\boldsymbol{V} \mid C_1 = 0)$, and similarly for the target domains.

[4]This assumption can be weakened further: in some circumstances one can infer from the data and the other assumptions that $C_1$ cannot have a direct effect on $Y$. For example: if there exists a descendant $D \in \text{DE}(Y)$, and if there exists a set $\boldsymbol{S} \subseteq \boldsymbol{V} \setminus (\{C_1, Y\} \cup \text{DE}(Y))$, such that $C_1 \perp\!\!\!\perp D \mid \boldsymbol{S}$, then $C_1$ is not a direct cause of $Y$ w.r.t. $\boldsymbol{V}$. For some proposals on alternative assumptions that can be made when this assumption is violated, see e.g., [Schölkopf et al., 2012, Zhang et al., 2013, 2015, Gong et al., 2016].

[5]This trivial observation is not novel; see e.g. [Ch. 7, p. 164, Spirtes et al., 2000]. It also follows as a special case of [Theorem 2, Pearl and Bareinboim, 2011]. The main novelty of this work is the proposed strategy to identify such separating sets.

[6]Note that this equation only holds asymptotically; for finite samples, in addition to the transfer from source domains to target domains, we have to deal with the generalization from empirical to population distributions and from the covariate shift if $\mathbb{P}(\boldsymbol{A} \mid C_1 = 1) \neq \mathbb{P}(\boldsymbol{A} \mid C_1 = 0)$ [see e.g. Mansour et al., 2009].

[7]Another option, proposed by Rojas-Carulla et al. [2018], is to *assume* that if $p(Y \mid \boldsymbol{A})$ is invariant across all source domains (i.e., $p(Y \mid \boldsymbol{A}, C_1 = 0, C_{\setminus 1} = c) = p(Y \mid \boldsymbol{A}, C_1 = 0)$ for all $c$), then the same holds across all source *and* target domains (i.e., $p(Y \mid \boldsymbol{A}, C_1 = 1) = p(Y \mid \boldsymbol{A}, C_1 = 0, C_{\setminus 1} = c)$ for all $c$). This assumption can be violated in some simple cases, e.g. see Example 2.

[8]Abstaining from predictions can be advantageous when trading off recall and precision. If a prediction *has* to be made, we can fall back on some other method or simply accept the risk that the transfer bias may be large.

[9]We build on the source code provided by Magliacane et al. [2016] which in turn extends the source code provided by Hyttinen et al. [2014]. The full source code of our implementation and the experiments is available online at `https://github.com/caus-am/dom_adapt`.

[10]Part of the CRM workshop on Statistical Causal Inference and Applications to Genetics, Montreal, Canada (2016). See also `http://www.crm.umontreal.ca/2016/Genetics16/competition_e.php`

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
