[Supplementary Material]

# Supplementary material for "Domain Adaptation by Using Causal Inference to Predict Invariant Conditional Distributions"

**Sara Magliacane**
MIT-IBM Watson AI Lab, IBM Research*
sara.magliacane@gmail.com

**Thijs van Ommen**
University of Amsterdam
thijsvanommen@gmail.com

**Tom Claassen**
Radboud University Nijmegen
tomc@cs.ru.nl

**Stephan Bongers**
University of Amsterdam
srbongers@gmail.com

**Philip Versteeg**
University of Amsterdam
p.j.j.p.versteeg@uva.nl

**Joris M. Mooij**
University of Amsterdam
j.m.mooij@uva.nl

## 1    A stronger version of the assumption in the main paper

In the main paper we make a non-standard assumption, Assumption 2(ii). Here we prove that this is assumption is a weakened version of two more standard assumptions, i.e. assuming the causal Markov and faithfulness assumptions in the source and target domains separately. Note that assuming these two assumptions instead of Assumption 2(ii) implies we cannot have perfect interventions in the target domain, which is otherwise allowed.

**Proposition 1.** *Assumption 2(ii), i.e. if $\boldsymbol{A} \cup \boldsymbol{B} \cup \boldsymbol{S}$ contains $Y$ but not $C_1$, then[2]*

$$\boldsymbol{A} \perp\!\!\!\perp \boldsymbol{B} \mid \boldsymbol{S} \, [C_1 = 0] \implies \boldsymbol{A} \perp\!\!\!\perp \boldsymbol{B} \mid \boldsymbol{S} \, [C_1 = 1];$$

*is implied by the following assumptions:*

*(a)  the pooled source domains distribution $\mathbb{P}(\boldsymbol{V} \mid C_1 = 0)$ is Markov and faithful to $\mathcal{G}^{\setminus C_1}$, and*
*(b)  the pooled target domains distribution $\mathbb{P}(\boldsymbol{V} \mid C_1 = 1)$ is Markov and faithful to $\mathcal{G}^{\setminus C_1}$,*

*where $\mathcal{G}^{\setminus C_1}$ denotes the induced subgraph of the causal graph $\mathcal{G}$ on the nodes $\mathcal{V} \setminus \{C_1\}$ (i.e., it is obtained by removing $C_1$ and all edges involving $C_1$ from the causal graph $\mathcal{G}$).*

**Proof.**  Let $\boldsymbol{A}, \boldsymbol{B}, \boldsymbol{S} \subseteq \boldsymbol{V} \setminus \{C_1\}$. By assumption, we have that

$$\boldsymbol{A} \perp\!\!\!\perp \boldsymbol{B} \mid \boldsymbol{S} \, [C_1 = c] \iff \boldsymbol{A} \perp \boldsymbol{B} \mid \boldsymbol{S} \, [\mathcal{G}^{\setminus C_1}]$$

holds for both $c = 0, 1$, which directly gives Assumption 2(ii).    □

## 2    Other proofs

**Proposition 2.**  *(Proposition 1 in the main paper) Under Assumption 2,*

$$\boldsymbol{A} \perp\!\!\!\perp \boldsymbol{B} \mid \boldsymbol{S} \, [C_1 = 0] \iff \boldsymbol{A} \perp\!\!\!\perp \boldsymbol{B} \mid \boldsymbol{S} \cup \{C_1\} \iff \boldsymbol{A} \perp \boldsymbol{B} \mid \boldsymbol{S} \cup \{C_1\} \, [\mathcal{G}]$$

*for subsets $\boldsymbol{A}, \boldsymbol{B}, \boldsymbol{S} \subseteq \boldsymbol{V}$ such that their union contains $Y$ but not $C_1$.*

**Proof.** First of all, $A \not\perp\!\!\!\perp B \mid S \ [C_1 = 0]$ implies (by definition) $A \not\perp\!\!\!\perp B \mid S \cup \{C_1\}$. Second, $A \perp\!\!\!\perp B \mid S \ [C_1 = 0]$ implies (by assumption) $A \perp\!\!\!\perp B \mid S \ [C_1 = 1]$, and taken together, we get $A \perp\!\!\!\perp B \mid S \cup \{C_1\}$. By the Markov and faithfulness assumption (Assumption 2(i)), this holds iff $A \perp\!\!\!\perp B \mid S \cup \{C_1\} \ [\mathcal{G}]$. $\qquad\square$

**Example 1.** *(Example 2 in the main paper) Assume that Assumptions 1 and 2 hold for two context variables $C_1, C_2$ and three system variables $X_1, X_2, X_3$ with $Y := X_2$. If the following conditional (in)dependences all hold in the source domains:*

$$C_2 \perp\!\!\!\perp X_2 \mid X_1 \ [C_1 = 0], \qquad C_2 \not\perp\!\!\!\perp X_2 \mid \emptyset \ [C_1 = 0], \qquad C_2 \perp\!\!\!\perp X_3 \mid X_2 \ [C_1 = 0], \qquad (1)$$

*then $C_1 \perp\!\!\!\perp X_2 \mid X_1 \ [\mathcal{G}]$, i.e., $\{X_1\}$ is a separating set for $C_1$ and $X_2$.*

**Proof.** In the JCI setting, we assume that in the full ADMG $\mathcal{G}$ over variables $\{C_1, C_2, X_1, X_2, X_3\}$, $C_1$ and $C_2$ are confounded and not caused by system variables $X_1, X_2, X_3$. Furthermore, no pair of system variable and context variables is confounded.

In the context $[C_1 = 0]$, if the conditional independences $C_2 \perp\!\!\!\perp X_2 \mid X_1 \ [C_1 = 0]$ and $C_2 \not\perp\!\!\!\perp X_2 \mid \emptyset \ [C_1 = 0]$ hold, then we can also derive that $C_2 \not\perp\!\!\!\perp X_1 \mid \emptyset \ [C_1 = 0]$, for example using Rule (9) from Magliacane et al. [2016]. Moreover, we know that $C_2$ is not caused by $X_1$ and $X_2$, or in other words $X_1 \not\dashrightarrow C_2$ and $X_2 \not\dashrightarrow C_2$. Thus we conclude that $(C_2, X_1, X_2)$ is an LCD triple [Cooper, 1997] in the context $C_1 = 0$. Since in addition, in this case $C_2$ and $X_1$ are unconfounded, the marginal ADMG $\mathcal{G}'$ on $\{C_2, X_1, X_2\}$ (in the context $C_1 = 0$, and hence by Proposition 1 in all contexts) must be given by Figure 1a.

Therefore, the extended marginal ADMG $\mathcal{G}''$ on variables $\{C_1, C_2, X_1, X_2\}$ must also have a directed path from $C_2$ to $X_1$ and from $X_1$ to $X_2$. $C_1$ cannot be on these paths, as none of the variables causes $C_1$, and therefore $\mathcal{G}''$ also contains the directed edges $C_2 \to X_1$ and $X_1 \to X_2$. Moreover, $\mathcal{G}''$ cannot contain any edge between $C_2$ and $X_2$, nor a bidirected edge between $X_1$ and $X_2$, because that would violate the conditional independence. By construction, in the JCI setting there is a bidirected edge between $C_1$ and $C_2$, and that is the only bidirected edge connecting to $C_1$ or $C_2$. As we assumed there is no direct effect of $C_1$ on target $X_2$, there is no edge between $C_1$ and $X_2$ in $\mathcal{G}''$. There is also no directed edge $X_1 \to C_1$ in $\mathcal{G}''$, as the JCI assumption implies none of the other variables causes $C_1$. Therefore, the marginal ADMG $\mathcal{G}''$ is given by Figure 1b, either with the directed edge $C_1 \to X_1$ present, or without that edge.

If it additionally holds that $C_2 \perp\!\!\!\perp X_3 \mid X_2 \ [C_1 = 0]$, we have two possibilities:

1. if $C_2 \perp\!\!\!\perp X_3 \mid \emptyset \ [C_1 = 0]$ holds, then $X_3$ is not caused by $C_2$. This means it cannot be on any directed path from $C_2$ to $X_1$, from $X_1$ to $X_2$, or be a descendant of $X_2$. Therefore the full ADMG $\mathcal{G}$ also necessarily contains the directed edges $C_2 \to X_1$ and $X_1 \to X_2$.

2. if $C_2 \not\perp\!\!\!\perp X_3 \mid \emptyset \ [C_1 = 0]$ holds, then in conjunction with $C_2 \perp\!\!\!\perp X_3 \mid X_2 \ [C_1 = 0]$ we can derive $X_2 \dashrightarrow X_3$, for example using Rule (5) from [Magliacane et al., 2016]. This means $X_3$ must be a descendant of $X_2$ in the full ADMG $\mathcal{G}$, which implies it cannot be on the directed path from $C_2$ to $X_1$, or on the one from $X_1$ to $X_2$. Therefore the full ADMG $\mathcal{G}$ also necessarily contains the directed edges $C_2 \to X_1$ and $X_1 \to X_2$.

(a) Marginal ADMG $\mathcal{G}'$.

(b) Set of candidate marginal ADMGs $\mathcal{G}''$.

(c) Set of candidate ADMGs $\mathcal{G}$.

Figure 1: ADMGs for proof of Example 1. Each dashed edge can either be present or absent.

Because of the independence statements and JCI assumptions, there cannot be a bidirected edge between $X_3$ and $X_1$, $X_2$, $C_1$ or $C_2$. Similarly, there cannot be directed edges from $X_3$ to one of those nodes. The edges $X_1 \rightarrow X_3$ and $C_2 \rightarrow X_3$ must also be absent.

In both cases, there can be a directed edge from $C_1$ to $X_3$. Therefore, the full ADMG $\mathcal{G}$ is of the form given in Figure 1c. In all cases we see that $C_1 \perp X_2 \mid X_1 [\mathcal{G}]$, and we conclude that $\{X_1\}$ is a valid separating set.

If the ADMG is as in Figure 2, then a standard feature selection method would asymptotically prefer the subset $\{X_1, X_3\}$ to predict $X_2$ over the subset $\{X_1\}$ (note that the Markov blanket of $X_2$ in context $[C_1 = 0]$ is $\{X_1, X_3\}$). As a result, any prediction method trained on all available features using source domain data (i.e., in context $[C_1 = 0]$) may incur a possibly unbounded prediction error when used to predict $X_2$ in the target domain $[C_1 = 1]$ (for example, if $X_3$ is an almost deterministic copy of $X_2$ if $C_1 = 0$, but has a drastically different distribution if $C_1 = 1$). $\qquad \square$

## 3   Additional results on synthetic data

We provide more information and experimental results for the synthetic data. We adapted the simulator of Hyttinen et al. [2014] to our setting. We generate randomly 200 acyclic models with three system variables, two context variables, and at most two latent variables (chosen randomly, so that the number of latent variables equals 1 or 2 each with probability $1/4$, and 0 otherwise). Each latent variable has two system variables as children, while the other variables have a random number of system variables as children, where system variables must be consistent with a chosen topological ordering, and where we enforce that a context variable may not simultaneously affect all system variables. The system and latent variables are each described by a linear structural equation with independent noise terms distributed as $\mathcal{N}(0, 0.0064)$. In these equations, each variable is multiplied by a coefficient sampled from $\mathcal{N}(0.2, 0.64)$ or $\mathcal{N}(-0.2, 0.64)$ (each with probability $1/2$ per variable). The context variables each correspond to an experimental domain; in their domain, that variable equals 1, otherwise it equals 0. This way, we simulate soft interventions. In order to scale the effect of these interventions, we multiply the coefficients of the context variables by the parameter $\gamma$, varying it from 0.1 to 100. We sample $N$ data points each for the observational and two experimental domains. Moreover, we randomly select $C_1$ and $Y$ from context and system variables respectively. We disallow direct effects of $C_1$ on $Y$.

As expected, our method performs well when the target distribution is significantly different from the source distributions. Figure 2 shows different settings with different scales of intervention effects. (In most graphs, the vertical axis has been adjusted to clearly show the boxplot, but leaving out the larger outliers.) In Figure 2a the intervention effects are all scaled by 0.1, resulting in very similar distributions in all domains. In this case, using our method does not offer any advantage with respect to the baseline and it actually performs worse. In the other cases, using our method starts to pay off in terms of prediction accuracy, and the difference increases with the scale of the interventions, as seen in Figure 2d.

In Figure 3, we vary the number of samples $N$ for each regime. The results improve with more samples, especially for our method, since the quality of the conditional independence test improves, but also for the baseline. In particular, as shown in Figure 3a, the accuracy is low for $N = 100$ samples, but it improves substantially with $N = 1000$ samples (Figure 2b).

## Footnotes

*Most of the work was performed while at the University of Amsterdam.

[2]Here, with $\boldsymbol{A} \perp\!\!\!\perp \boldsymbol{B} \mid \boldsymbol{S} \, [C_1 = 0]$ we mean $\boldsymbol{A} \perp\!\!\!\perp \boldsymbol{B} \mid \boldsymbol{S} \, [\mathbb{P}(\boldsymbol{V} \mid C_1 = 0)]$, i.e., the conditional independence of $\boldsymbol{A}$ from $\boldsymbol{B}$ given $\boldsymbol{S}$ in the mixture of the source domains $\mathbb{P}(\boldsymbol{V} \mid C_1 = 0)$, and similarly for the target domains.

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

(a) Synthetic data with a small perturbation ($\gamma =$ 0.1) and $N = 1000$ samples.

(b) Synthetic data with a medium perturbation ($\gamma = 1$) and $N = 1000$ samples.

(c) Synthetic data with a large perturbation ($\gamma =$ 10) and $N = 1000$ samples.

(d) Synthetic data with a very large perturbation ($\gamma = 100$) and $N = 1000$ samples.

Figure 2: Additional results when varying the causal effect of all interventions ($\gamma$).

(a) Synthetic data with $N = 100$ samples per regime and a large perturbation ($\gamma = 10$).

(b) Synthetic data with $N = 1000$ samples per regime and a large perturbation ($\gamma = 10$).

(c) Synthetic data with $N = 5000$ samples per regime and a large perturbation ($\gamma = 10$).

Figure 3: Additional results when varying the sample size per regime ($N$).