[Reviews · NeurIPS 2018]

Reviewer 1



Summary: this paper proposes a new approach to domain adaptation that relies on the identification of a separating feature set, conditional on which the distribution of a variable of interest is invariant under a certain intervention. The problem that the authors set out to solve is a vey ambitious and difficult one. I think that they are partially successful in their attempt, although I have a number of serious reservations. Detailed comments follow: 1) I found the paper generally very well written and clear. I come from a `potential outcomes’ school of causal inference, and my knowledge of DAGs is not extensive — even so, the authors made a reasonable effort to introduce all needed concepts and I was able to follow the details of the exposition. I believe this article to be accessible to most researchers who have a solid background in causal inference, even if their knowledge of DAGs is limited. 2) I appreciate the fact that the authors (mostly) avoided making grandiose claims about the reach of their method. They acknowledge some of its limitations openly. I also greatly appreciate the innate conservativeness of their algorithm (if we accept their assumption) which may refuse to make predictions if it cannot find a separating feature set. 3) If we accept the assumptions made by the authors, then although the basic idea is not groundbreaking (the authors acknowledge as much in footnote 4 on page 5), their contribution to identifying separating sets of features is original and interesting. 3) My first serious criticism is with the assumptions being made. First of all, I think that more space should have been devoted to discussing the assumptions. I understand that 10 pages doesn’t leave a lot of space, but the entire premise of this work rests on these assumptions. The Markov and Faithfulness assumptions being standard, it’s ok to not expand on them too much (although it never hurts to discuss an assumption). Assumptions 1 and 2 are another matter entirely — neither is standard and both are crucial to the rest of the paper. While I am roughly ok with Assumption 1, I have serious reservations about Assumption 2. Specifically, Assumption 2(ii) and Assumption 2(iii) seem to be assuming away most of the problem. At a minimum, the authors should provide (more than one) concrete example where both assumptions are credible. My main contention is that 2(ii) and 2(iii) assume a strong prior knowledge of the causal mechanisms in the system, and I cannot envision a scenario in which an individual with such knowledge would still have a need for domain adaptation. 4) My second serious criticism is with the practical aspect of this method. The authors acknowledge the computational complexity of the approach, but don’t discuss the details of this limitation. Concretely, how many variables can this method handle? Both examples in Section 3 deal with very few variables, far fewer than what would be required in most practical scenarios I can think of. The authors state in Section 4 that “more work remains to be done on the implementation side […] scaling up to more variables”. Do you have a sense for what the algorithmic complexity of the problem is? How likely is it that any significant scaling improvements (say handling ~50 variables) will occur soon? Do you have concrete ideas about how to get there? Additional minor comments: 1) The first sentence of Example 1 is awkward. It’s a subordinate clause that isn’t attached to a main clause. 2) Same comment for the first sentence of Task 1. 3) I found Example 1 to be unclear. You should make it clear earlier that the example assumes $P(Y | X_1, C_1 = 0) = P(Y | X_1, C_1=1)$ instead of writing it at the end. 4) The labels of the plots in Figure 3 are way too small.

Reviewer 2



The authors propose a novel approach to domain adaption the relaxes many of the standard assumptions. In particular, they address the setting when both the causal graph and the intervention types and targets are unknown. Their methodology follows two steps: first, they select a set of features which are likely to be invariant; second, they use these features to draw causal conclusions. Although I am not very familiar with this branch of causal inference, I found the paper well written and easy to follow. I have some minor comments listed below: - Connect the examples to real-world applications. - Typo line 109 "Given are data" - Fix the axis in your figures so that they are they can be read without having to zoom in. - What happens if one or more of the conditions in Assumption 1 do not hold? - Are there any strategies to asses the validity of Assumption 1?

Reviewer 3



Given observational data (on all variables) from source domain(s) and interventional data (on all variables except Y) from target domain(s), the paper develops a technique to predict P(Y). The paper relies primarily on a certain conditional independence assumption between context and system variables. Causal graph is partially constructed to check if this assumption holds. Solution is inspired by ideas from varied areas such as graphical models, causal structure learning and domain adaptation. The advantages of this work over existing work are that it does not require (i) causal graphs as input, (ii) interventions to be perfect and (iii) targets of interventions to be known. The problem and assumptions have been very clearly stated. Related work is written well. Example 1: It would be helpful if you additionally mention that in this case intervention targets are known. Also on line 64, where you note that observation measurements are available, it will be helpful if you state the same using notations as well. Line 35: An informal definition of imperfect interventions will be helpful.